# 3D Stereophotogrammetric Quantitative Evaluation of Posture and Spine Proprioception in Subacute and Chronic Nonspecific Low Back Pain

**DOI:** 10.3390/jcm11030546

**Published:** 2022-01-22

**Authors:** Edyta Kinel, Piero Roncoletta, Tiziana Pietrangelo, Moreno D’Amico

**Affiliations:** 1Chair of Rehabilitation and Physiotherapy, Department of Rehabilitation, University of Medical Sciences, 61-701 Poznan, Poland; 2SMART (Skeleton Movement Analysis and Advanced Rehabilitation Technologies) LAB, Bioengineering & Biomedicine Company Srl, 66020 San Giovanni Teatino, Italy; bbcmail@tin.it (P.R.); damicomoreno@gmail.com (M.D.); 3Department of Neuroscience, Imaging and Clinical Sciences, University “G.d’Annunzio”, Chieti-Pescara, 66100 Chieti, Italy; tiziana.pietrangelo@unich.it

**Keywords:** nonspecific low back pain, stereophotogrammetry, posture, spine, spine deformities, self-correction, proprioception

## Abstract

The literature shows that low back pain causes a reduced lumbar range of movement, affecting patients’ proprioception and motor control. Nevertheless, studies have found that proprioception and motor control of the spine and posture are vague and individually expressed even in healthy young adults. This study aimed to investigate the standing posture and its modifications induced by an instinctive self-correction manoeuvre in subacute and chronic nonspecific low back pain (NSLBP) patients to clarify how NSLBP relates to body upright posture, proprioception, and motor control and how these are modified in patients compared to healthy young adults (121 healthy young adults: 57 females and 64 males). A cohort of 83 NSLBP patients (43 females, 40 males) were recruited in a cross-sectional observational study. Patients’ entire body posture, including 3D spine shape reconstruction, was measured using a non-ionising 3D optoelectronic stereophotogrammetric approach. Thirteen quantitative biomechanical parameters describing the nature of body posture were computed. The statistical analysis was performed using multivariate methods. NSLBP patients did not present an altered proprioception and motor control ability compared to healthy young adults. Furthermore, as for healthy subjects, NSLBP patients could not focus and control their posture globally. Proprioception and motor control in natural erect standing are vague for most people regardless of gender and concurrent nonspecific low back pain. Self-correction manoeuvres improving body posture and spine shape must be learned with specific postural training focusing on the lumbar spine.

## 1. Introduction

The problem of “low back pain” (LBP) involves an ever-increasing percentage of the population, with the potential for further high growth due to the progressive increase in the average age [1,2,3,4]. For nearly all people presenting LBP, the specific nociceptive source cannot be identified, and those affected are then classified as having so-called nonspecific LBP (NSLBP) [2,4,5]. The World Health Organisation (WHO) report indicates that, throughout life, NSLBP patients have a high prevalence worldwide, reaching 60–70% in developed countries [6], with a strong economic and social effect being the leading cause of activity limitation and absence from work worldwide [2,4,5,7,8,9].

Recent guidelines have put minor emphasis on pharmacological and surgical treatments [3,4,5], while they promote postural self-awareness through exercises to provide the patients with the proper physical and proprioceptive tools for an early return to activity as the first-line treatment [3,10].

Proprioception, also referred to as kinaesthesia, is the sense of self-movement and body position. It enables us to perceive the location, movement, and action of parts of the body while providing the perception of muscle force and effort, heaviness, stiffness, and viscosity. It encompasses a complex of sensations that arise from signals of sensory receptors, the proprioceptors, i.e., mechanosensory neurons located within muscles, tendons, skin, and joints, and from central signals related to motor output. Proprioceptive signals are transmitted to the central nervous system, where they are integrated with information from other sensory systems, such as the visual system and the vestibular system, to locate external objects relative to the body and contribute to creating an overall representation of body position, movement, and acceleration. Proprioception is closely tied to the control of movement [11,12].

Motor control is defined as how the nervous system regulates posture and movement to perform a specific motor action, considering all related motor, sensory, and integrative processes [11,12].

In the last few years, the connection between altered motor function and LBP has been explored. An entire book has been devoted to openly and intensely discussing such a topic [13]. Recent reviews highlight that adaptions of motor control likely play a significant role in chronic or recurrent LBP, as there is evidence that potential loss of trunk control and enhanced trunk muscle co-contraction could induce increased spinal tissue strains resulting in muscle fatigue contributing to LBP chronicisation [14,15,16,17]. Both factors, i.e., lack of trunk control and increased muscle co-contraction, have been related to prolonged mechanical loading on spinal tissues, potentially accelerating intervertebral disc and other tissue degeneration [18,19].

Pain causes a redistribution of activity, which contributes to changing mechanical behaviour. While these changes produce immediate benefits, they can lead to adverse effects in the long run [17,20,21,22].

However, altered postural regulation cannot be clarified by simple variations in motor system excitability. Changes in the motor system occur at different stages. It is hypothesised that changes in the nervous system in response to pain cause changes in the neuromuscular system, which can have a complementary, additive, or competitive effect. Remodulation on neuromuscular activity can involve the motor cortex. Meier et al. [22], in their review, showed the potential contribution of supraspinal motor control and proprioception to the changes induced by pain. They concluded that “back pain-induced disrupted or reduced proprioceptive signalling likely plays a pivotal role in driving long-term changes in the top-down control of the motor system via motor and sensory cortical reorganisation”.

On the basis of such research findings, a rehabilitation exercise approach was developed named “motor control exercise” (MCE) in LBP. MCE is used to describe techniques that use exercises that aim to alter how an individual controls their body (i.e., posture/alignment, movement, and muscle activation) to modulate the loading of the spine and adjacent structures [13,21].

Despite its increased popularity in research and clinical practice over the past two decades, MCE for LBP did not produce clear evidence. Several systematic reviews focused on the effectiveness of MCE [16,23,24]. While it has been confirmed that MCE is better than minimal intervention in reducing pain in the short, intermediate, and long term, and in decreasing disability at long-term follow-up, contrasting evidence for the comparison of effects of MCE compared to other interventions has been reported in the literature [16,23]. Such inconsistency found in the reviews has been explained likely by studies taking a one-size-fits-all approach, even though the literature shows that patients’ characteristics vary, as do motor control pain-induced modifications [21].

Summarising, it can be said that LBP induces changes in all levels of sensorimotor control and, therefore, also in upright posture. Similarly, it can be assumed that incorrect posture may be among the causes that determine the onset of LBP and/or its chronicity. Surely the composition of both families of phenomena, possibly interacting, is declined in a strictly individual way depending on the starting postural condition.

It has been demonstrated that to be asymptomatic in healthy individuals does not mean to have an optimal posture [25].

Furthermore, studying the postural changes during the instinctive self-correction manoeuvre (ISCO), D’Amico et al. (2018) [26] found that proprioception and motor control of posture and spine in healthy young adults are generally vague and individually expressed.

A recent study confirmed the association between poor proprioception and motor control and nonoptimal upright posture and spine balance by analysing the ISCO capabilities of adolescents with idiopathic scoliosis. Surprisingly, it has also been discovered that, while the posture of scoliotic patients was worse than that of healthy young adults, conversely, no significant differences were found in ISCO performances between the two populations [27].

In these two latter studies, the ISCO manoeuvre was used to characterise the proprioceptive abilities in a standing posture to overcome the drawbacks of other methodologies previously adopted in the literature. Indeed, most biomechanical quantitative studies on proprioception in healthy individuals and people with LBP have focused on repositioning accuracy in tests of spinal position sense [28]. However, it has been highlighted that these tests imply several repetitions. This condition could represent a problem with this type of testing because acuity during repeated testing can be improved due to a learning effect [28].

Therefore, it was suggested that the test should represent the postural challenges that the body must continually respond to under changing circumstances in common everyday conditions. Testing protocols that assess initial responses might be more relevant to the ‘real world’ and may better demonstrate impairments in people with low back pain [28]. In such a context, neuromuscular activation patterns and the effect on biomechanical outcome parameters have to be considered to quantitatively evaluate postural control during standing [17].

Thus, the choice of the ISCO manoeuvre showed multiple advantages: (1) it does not involve a learning effect; (2) it involves the entire top-down chain of motor control (cerebral cortex, proprioception, somatosensory perception, muscle effectors, and motor output expressed by CNS); (3) it is associated with the internal calibration and the internal image [29,30] that each subject builds up about what their correct posture should be.

Considering all the above, it is not clear whether modified proprioception and reduced motor control determine a poor posture and the onset of LBP, whether the presence of LBP reduces proprioceptive and motor control abilities, or whether a vicious cycle of mutual influence leading to the onset and chronicisation of LBP is triggered. This uncertainty may also be related to the reduced availability of measurement methods to provide a complete quantitative posture description.

A few studies in the literature analysed the standing posture of subjects with LBP, but just a few postural parameters were evaluated [17]. A recent study on the LBP population [31] considered the complete postural description using the stereophotogrammetric approach [25,26,32,33,34], including assessing the 3D morphology of the spine. The research confirmed a direct link between bad standing posture associated with spine deformities and LBP. Specifically, it was demonstrated that patients with chronic and subchronic LBP have a more unbalanced posture associated with more significant spinal deformities and asymmetries of underfoot loads than healthy young adults. These postural conditions were closely related to the presence of a leg length discrepancy (LLD). The study showed that postural rebalancing induced by LLD equalisation led to total pain remission associated with the reduction of spinal deformities and the symmetrisation of underfoot loads.

The current research aimed to investigate the standing posture and its modifications induced by an ISCO manoeuvre in chronic and subacute NSLBP patients to clarify how NSLBP is connected with body upright posture, proprioception, and motor control. To this purpose, performances of NSLBP patients in the standing posture and ISCO manoeuvre were compared with those of healthy young adults using the non-ionising real-time optoelectronic stereophotogrammetric measuring method introduced by [25,26,32,33].

The comparison was carried out separately for males and females to consider if proprioception and motor control are gender-dependent in NSLBP patients.

## 2. Materials and Methods

### 2.1. Study Design

The present study is a prospective, cross-sectional observational investigation (according to the STROBE guidelines [35] and the Helsinki Declaration), evaluating 3D quantitative posture and spine proprioceptive perception through an instinctive self-correction manoeuvre in nonspecific chronic and subacute low back pain patients.

The Ethics Committee of the University of Medical Sciences in Poznan, Poland, approved this study (resolution number: 374/17). All participants signed a written informed consent form.

Data collection took place between May 2017 and December 2019.

### 2.2. Participants

Before the measurement session, participants were given a thorough clinical postural examination by an experienced physiotherapist (the first author). Before evaluation, pain intensity was rated via the numerical rating scale (NRS) [36].

The inclusion/exclusion criteria were as follows: diagnosis of nonspecific low back pain; subacute (≥6 weeks) and chronic (≥12 weeks) nonspecific lumbar pain history; males and females 20–40 years old (Caucasian); no neurologic problems; no history of musculoskeletal system injury or surgery; no physiotherapeutic exercise treatment of any kind in the last 6 months (to reduce confounding effects in specific motor control capabilities induced by exercise training).

A cohort of 83 NSLBP volunteer patients (40 males, 43 females) was recruited at the Clinic of Rehabilitation, University of Medical Sciences, Poznan-Poland.

The performances of such NSLBP patients in indifferent orthostasis (IO) and instinctive self-correction manoeuvre (ISCO) were compared to those of 121 healthy young adults (57 females and 64 males) selected in previously published research [26]. NSLBP and healthy young adults’ characteristics are summarised in Table 1 and Table 2.

Comparing NSLBP patients’ characteristics to those of the reference healthy young adults, NSLBP patients resulted on average 5–7 years older than healthy young adults, with a statistically significant greater weight (males *p* = 0.0097, females *p* = 0.0011) and body mass index (BMI) (males *p* = 1.9 × 10^−6^, females *p* = 0.00013). In any case, BMI was still within the normality range for both genders [37].

### 2.3. Data Measurement

Our experimental recordings were based on a 6TV cameras GOALS Global Opto-electronic Approach for Locomotion & Spine) (Bioengineering & Biomedicine Company S.r.l. Pescara-Italy) (stereophotogrammetric opto-electronic system derived from the Optitrack System (NaturalPoint Inc., Corvallis, OR, USA) (resolution 1.3 MP, 120 fps, error range 0.3 mm, calibrated volume 3 × 3 × 2 m [25,26,33]). One synchronous baropodometric platform (Zebris FDM-SX) (Zebris Gmbh, Isny, Germany) (active area dimensions: 400 mm × 330 mm, total 1920 square capacitive sensors; 1.4 sensors/cm^2^) was used to measure bilateral foot pressure maps and underfoot vertical forces exerted on each foot in standing position (Figure 1). The platform is rectangular, and the sensors are arranged in rows and columns parallel to the shorter and longer edge, respectively. The manufacturer grants an accuracy of the calibrated pressure measuring range (1–120 N/cm^2^) of ±5% of the maximum range. An essential step of the calibration procedure is to establish the relative position of the baropodometric platform within the calibrated volume. This position is necessary to proceed to all subsequent calculations related to underfoot pressure maps and associated vertical forces. The ground plane is established on the baropodometric platform surface during the stereophotogrammetric calibration step once such a surface is adjusted to be perfectly horizontal [25,32]. The calibration procedure defines the laboratory reference system. Its origin is placed on the top left rectangular platform vertex. The *X*- and *Z*-axes are defined along platform edges, with the *X*-axis along the shorter edge and the *Z*-axis along the longer edge. Following the righthand rule, the *Y*-axis is orthogonal to the ground plane (i.e., in the direction of the gravity line).

The 27 body landmark protocol, labelled by passive retroreflective markers [38], was used to measure the subjects’ 3D whole skeleton posture. The model’s accuracy and precision were founded on in-house original signal processing and optimisation procedures [39,40,41,42,43,44,45,46] and anatomical studies listed in the literature (cadaver dissections, in vivo X-ray, and parametric regression equations from gamma-ray measurements) [47,48,49,50]. The model was formulated in a parametric form to scale any subject’s characteristics by fitting each given skeletal segment to the 3D measured positions of its corresponding body landmarks [39]. The model was tested extensively in the clinical environment to analyse human posture (Figure 2) [25,26,32,33,51]. The software package called ASAP 3D Skeleton Model (Bioengineering & Biomedicine Company S.r.l. Pescara-Italy), implementing a full 3D parametric biomechanical human skeleton model (3D spine included), was used to carry out data processing.

### 2.4. Acquisition Protocol

The standard trial session aimed to define the participant’s indifferent orthostasis (IO) (i.e., maintaining the most natural erect posture) and, soon after, their instinctive self-corrected orthostasis by giving the patient a generic command, i.e., the request to assume the best correct self-perceived standing posture without adding any specific indication. After each ISCO manoeuvre measurement, the subject was left to relax a few seconds before the next ISCO manoeuvre performance. Even if the ISCO manoeuvre was performed five consecutive times, no learning effect could be hypothesised given that no information was provided to the patient about their performance. Worth underlining, the ISCO manoeuvre did not elicit any additional pain to any subject, and no one reported to be somewhat conditioned by pain during the ISCO manoeuvre performance.

The same commands were given in [26] when healthy young adults were analysed. All measurements were taken between 12:00 and 7:00 p.m. to reduce potential postural effects resulting from circadian rhythms. The subjects were asked to avoid intensive training and/or demanding physical activity before the postural assessment. The assessment/measurement session aimed to fully capture and record the subject’s neutral standing posture, with the upper arms relaxed along the side of the body and eyes looking directly ahead in the horizontal plane. Such posture has been defined as indifferent orthostasis (IO). For all subjects, marker positioning was performed by a single operator with more than 20 years of experience. Each marker-positioning session lasted approximately 10 min, after which the subject was asked to sit for a few minutes. Afterwards, the individual was asked to keep an indifferent orthostasis standing with both feet on the baropodometric platform [25,33].

Different positions of the feet can influence IO and ISCO postures. Thus, the subject was asked to align heels on a line parallel to the frontal plane (i.e., on a line parallel to the *X*-axis of the laboratory reference system) and keep feet apart at about pelvis width (i.e., with feet under the hip-joint projection), without restricting foot directions, to avoid foot position influence. Real-time baropodometric measurement availability allows controlling foot alignment straightforwardly by checking that the heels’ most prominent tip lay on the same row in the foot pressure maps. The subject was positioned in front of a blank wall to avoid visual feedback or reference during measurements.

At least five subsequent acquisitions lasting 2 s at a sampling rate of 120 Hz were recorded for each IO and ISCO attitude. Thus, a total of 1200 3D measurements for each static posture were averaged. Before averaging, a level of pre-processing is needed on the acquired 3D raw data to define the subject’s local coordinate system and its orientation relative to the global coordinate system [25,26,33,39,40]. We used the general definitions provided by the Scoliosis Research Society [52]. However, in distinction to such recommendations, PSIS rather than ASIS landmarks were considered in defining the subject’s local coordinate system to reduce propagation errors and/or other interference deriving from pelvis torsion in the subsequent calculation of spinal parameters. Hence, in a righthanded system: the frontal coronal (*YZ*) plane is the vertical plane containing the PSIS. The *Z*-axis is the vertical axis pointing up from the mid-point between the PSIS in such a plane. The *Y*-axis is orthogonal to the *Z*-axis, passing through the mid-point between the PSIS, pointing from the body’s right side to the left. The *X*-axis is determined by the righthand rule passing through the mid-point between the PSIS pointing forward, i.e., from the back to the body’s front. The *XZ*-axes define the sagittal plane; the *XY*-axes define the horizontal plane [25]. Once the subject’s local coordinate system is defined, its origin is translated into the S3 position for anatomical convenience. Once having determined this individual system, a rotation is performed within each frame to align the subject’s coordinates with the global reference coordinates. When the alignment is complete, it is possible to properly average all acquired frames. On the basis of the 11 3D spinous processes measurements, data are interpolated using cubic splines to assess the position of each unlabelled spinous process and intervertebral disc. After interpolation, the space curve modelling of the spine is analytically represented using three parametric functions *x*(*t*), *y*(*t*), and *z*(*t*) (with the parameter being *t* > 0). A smoothing and differentiation procedure specially developed for interpolated data with cubic splines is applied to these functions [39,43,44,45]. Once the three parametric functions *x*(*t*), *y*(*t*), and *z*(*t*) and their derivatives are assessed, the 3D position of each vertebra from C7 down to S3 is derived. The maxima and minima of the assessed first derivative allow selecting, under analytical constraint, all and only the inflexion points defining the limit vertebrae. After determining the limit vertebrae, the Cobb and kypho-lordotic angle computation is straightforward, computing the angle between the tangents at such points [25,32,33,39]. The introduced numerical processing technique showed a maximal error of less than 1° (approximately) for the Cobb computed angle on a curve of about 65°, modelled with a simulated sinusoidal data series with superimposed white noise (σ = 1 mm) [39].

We decided to consider the Cobb angle value of the two major curves (CA1, CA2, Table 2) for statistical analysis regarding the spinal deformities in the frontal plane.

Figure 3 and Figure 4 show an example of data elaboration outcome and the related graphical report of the IO vs. ISCO measurement comparison in the frontal and sagittal planes, respectively. A video showing the acquisition/elaboration processes can be found in the Appendix A.

A set of 13 significant parameters detailing the three-dimensional body posture structure were computed from the 3D biomechanical human skeleton model reconstruction [25,26]. Such variables were subdivided into three groups, as reported in Table 3.

### 2.5. Group Statistical Analysis

The statistical analysis to compare female vs. male groups in IO and ISCO, the by gender IO vs. ISCO, and the by gender NSLBP vs. healthy young adults was performed using a multivariate method, provided the checked correlation (through the computation of correlation matrices) among the considered 13 quantitative postural parameters. The paired samples Hotelling’s T^2^ test was applied in the IO vs. ISCO comparisons. Conversely, for the female vs. male in IO and ISCO and the by gender NSLBP vs. healthy young adults, the independent samples Hotelling’s T^2^ test was applied. The simultaneous 95% confidence intervals (derived from Hotelling’s tests) were undertaken to determine the statistical significance of the difference of means for each of the 13 quantitative parameters [53]. Such a method is preferable compared to setting a battery of separate *t*-tests for each variable with Bonferroni correction on the type I error (α’ = α/k) because the latter approach does not take into account the correlation between the variables and, therefore, results in an overcorrection of the significance value α [53].

The comparison, female vs. male, was performed in IO to analyse eventual postural gender differences and subsequently in ISCO to investigate an eventual different self-correction ability by gender. Comparing NSLBP patients vs. healthy young adults, both in IO and ISCO, allows highlighting eventual postural, proprioception, and motor control differences between the two groups.

### 2.6. Intra-Subject Statistical Analysis

At the intra-subject level, we investigated how the ISCO modified the subject’s posture by improving, worsening, or unchanging the original attitude. The comparison was performed through a *t*-test between the mean values of 13 considered quantitative parameters obtained per participant in the IO and the ISCO postures.

The actual postural parameter was classified as “*unchanged*” if there was no statistically significant difference.

Conversely, we defined the following as “*improvement*”:Frontal plane parameters: when the parameter values approached the optimal theoretical zero value during the ISCO [26].Sagittal plane parameters: in this case (except for pelvis torsion (|PT|) which should be zero), there are no theoretical optimal reference values; hence, we decided to consider the normative data determined in previous studies in healthy young adults, for IO and ISCO, as reference values to be approached [25,26,33].|∆UL| (i.e., the difference in underfoot load between the feet): the optimal theoretical condition is achieved when there is a perfect balance of underfoot load distribution between the left and right sides; therefore, there was “*improvement*” when changes approached this condition.

“*Worsening*”: each time, during the self-correction manoeuvre (ISCO), that a statistically significant change differed from the definitions of “*improvement*”, it was concluded that a “*worsening*” occurred.

### 2.7. Summarising Indices

A summarising index was defined for each patient, assigning +1, −1, or 0 scores when an improvement, a worsening, or no change was respectively determined [25,26]. Henceforth, a “global postural index” (GPI_i_) given by the sum of scores obtained for all the variables for the ith participant was defined [25,26]. The Frontal Plane Index (FPI_i_) and the Sagittal Plane Index (SPI_i_) were defined by the sum of scores for the variables of the related group [25,26] (Table 3).

Each of the summarising indices was regarded as an “*improvement*” if the summed parameters got a positive score ≥50% of the maximum obtainable positive score; conversely, it was regarded as “*worsening*” if such a sum got a negative score ≥50% of the maximum obtainable negative score and “*unchanged*” in the other cases [25,26].

By counting the number of “*improvement*”, “*worsening*”, and “*unchanged*” indices obtained for each participant in each parameter, it was possible to determine the percentages of “*improvement*”, “*worsening*”, and “*unchanged*” achieved in the male and female subgroups.

### 2.8. Power Analysis and Sample Size

Hotelling’s T^2^ independent test gives the most critical condition for NSLBP patients, among the various multivariate comparison tests, when males vs. females are compared in IO and ISCO. According to GPower software [54], 42 patients per each gender are necessary to grant an effect size d = 1.00 (Mahalanobis distance), fixing the required power = 80%, α = 5%, and k = 13 (number of variates). Unfortunately, two male patients abandoned the trial. In any case, the final sample of 40 males and 43 females granted an effect size d = 1.0019.

Conversely, for Hotelling’s T^2^ paired version (IO vs. ISCO in NSLBP patients), d = 0.80 for the male group and d = 0.76 for the female group were granted.

## 3. Results

### 3.1. Group Statistical Analysis

We investigated gender differences in any considered postural parameters in group statistical analysis, both in IO and in ISCO. As shown in Table 4, only lumbar lordosis angle (LLA), sacral angle (SA), and sagittal averaged spinal offset (ASO SG) were different in both orthostatic conditions. Conversely, underfoot load discrepancy (|∆UL|) resulted statistically different between genders only in IO. In ISCO, the males improved their balance, reducing the underfoot load asymmetry and approaching the female value that remained stable, while keeping the same underfoot load asymmetry determined in IO.

The IO vs. ISCO comparison (Table 5) demonstrated that NSLBP patients, both sexes, were able to modify their posture only regarding the following parameters: thoracic kyphosis angle (TKA) and sacral angle (SA). Females modified their sagittal averaged spinal offset (ASO SG) in ISCO, generating a worse forward trunk leaning than IO.

Table 6 reports the list of statistically different parameters in IO or ISCO of NSLBP patients vs. healthy young adults. For females, only two parameters appeared to be different between NSLBP patients and healthy young adults in both considered postures, i.e., pelvis obliquity (|∆PSIS|) and primary Cobb angle (CA1). Frontal averaged global offset (|AGO FR|) was different in IO, while secondary Cobb angle (CA2) was different in ISCO.

NSLBP males showed a different behaviour to that described for females in that they presented a higher number of differences than healthy young adults. Indeed, five parameters were different in both IO and ISCO (|∆ASIS|, |∆PSIS|, ASO SG, AGO SG, and |∆UL|); SA differed only in IO, while CA1 and CA2 were significantly greater than those of healthy young adults in ISCO.

Evaluating the principal curve position along the spine, expressed by CA1, we found that 37.5% of males and 37.2% of females had the primary curve in the thoracic region, while 27.5% and 21.0% had the primary curve in the lumbar, and 35.0% and 41.8% had the primary curve in the thoracolumbar regions for males and females, respectively.

### 3.2. Intra-Subject Statistical Analysis

The intra-subject level analysis confirmed that participants presented a generalised difficulty in modifying their posture (Table 7). The unchanged percentage was predominant; GPI ranged from 80% in males to 93% in females. However, changes were present at the level of some single parameters (TKA, |∆UL|, and |PT| for both genders; SA and |∆ASIS| in males), mainly in the sagittal plane. About two-thirds of the total NSLBP patients presented no change in LLA values. In those patients who showed changes in ISCO, males performed better than females globally (GPI) and in both the frontal (FPI) and the sagittal plane (SPI), while females performed better in underfoot load rebalancing (|∆UL|).

## 4. Discussion

Our research aimed to study the proprioceptive abilities of NSLBP patients, compared to those of healthy young adults, by assessing postural changes during the instinctive self-correction manoeuvre (ISCO). Furthermore, the study aimed to characterise, by gender, the posture and spine morphology of NSLBP patients. The ISCO manoeuvre involves the entire top-down chain of motor control: cerebral cortex, somatosensory perception, muscle effectors, and motor output expressed by CNS. This manoeuvre is also associated with the internal calibration and the internal image [29,30] that each subject builds up about what their correct posture should be.

The study was performed through the advanced non-ionising real-time optoelectronic stereophotogrammetric measuring method [32] that proved to be a very accurate and detailed solution in 3D posture analysis and self-correction measurement on a healthy young adult population [26].

The study assumed that, if during ISCO, a subject improves their posture compared to IO, this is a suitable indication of their appropriate instinctive proprioceptive perception and good motor control. “For optimal postural control, the CNS must identify and selectively focus on the sensory inputs (visual, vestibular, and proprioceptive) providing the most functionally reliable input. Presumably, reliability is determined by some kind of comparison with internal representations contained in cortical, subcortical, and cerebellar body maps” [28].

Regarding IO posture characteristics, in comparing NSLBP patients vs. healthy young adults, males and females presented slightly different behaviours. In any case, patients exhibited a generally worse posture, showing an increased pelvis obliquity, being more unbalanced (i.e., side and back/forward leaning with respect to gravity line, underfoot loading discrepancy), and presenting higher spinal deformities. Specifically, NSLBP males resulted backward unbalanced, with pelvis retroversion and higher underfoot load discrepancy than healthy young adults. Similar results were found in D’Amico et al. (2021) [31].

Results showed no gender differences in NSLBP patients in the ISCO manoeuvre’s performance except for a greater forward trunk leaning (and, thus, a sagittal trunk attitude worsening) achieved by females vs. males.

Except for the above-summarised essential anatomical–structural difference found in the NSLBP sample, patients did not present an altered proprioception and motor control ability compared to healthy young adults.

The following behaviours previously recognised in healthy young adults were found in the NSLBP group: self-correction manoeuvre leading to a global improvement of standing natural erect posture was not instinctive, but had to be learned with specific postural training; subjects could not focus and control their posture in a global way, but only in a few aspects at a time; participants showed better “attention” to the sagittal plane but with substantial neglect of the spine’s lumbar tract; the postural changes of the ISCO manoeuvre rarely induced a better posture than IO.

When evaluating the ISCO performance, we refer to the so-called “ideal posture”, which should express the “optimal” biomechanical and neurophysiologic functions. Unfortunately, there is not a “gold standard” defining “ideal posture” as individual authors have different views of “ideal posture” [55]. Posture is an open solution in human beings; therefore, even considering the natural standing posture, several “optimal” solutions can be identified, depending on adaptation to particular functions. Indeed, it is well known that, in some sports, practitioners or even elite athletes develop some level of hyperlordosis or hyperkyphosis or even trunk asymmetries [56,57,58] that could be considered as the result of a long period of training and adaptations, thus becoming, in some way, “optimal” for the requested sports motor tasks. However, such specialised optimisation does not necessarily preserve subjects from presenting posture or spine problems [56,58].

It is commonly accepted that the best theoretical solution for frontal plane posture is to achieve symmetry in balance and loads [55,59]. However, for the sagittal plane posture, the optimal value has not yet been defined. Indeed, it represents an open debate in the literature. Thus, the most reasonable reference value to consider is the normative data determined in a healthy young adult population [25].

Postural control, defined as the control of the body’s position in space for balance and orientation in static and dynamic conditions [60], results from the complex interplay of somatosensory perception and motor control expressed by the CNS, i.e., from a constant interaction between motor outputs to effectors (e.g., paraspinal muscles) and sensory inputs (e.g., proprioception) on various levels of the nervous system [12,13,61]. The proprioceptive system provides information on joint angles, as well as changes in joint angles, joint position, and muscle length and tension, while the tactile system is associated mainly with sensations of touch, pressure, and vibration [13,62]. Therefore, visual, vestibular, and somatosensory modalities are involved in balancing and orienting the body in the vertical position when standing. Thus, as the posture is the outcome of integrating all the described components, it is not easy to separate each sensory perception contribution unless well-designed experiments are set. Nonetheless, in our research, we measured the most instinctive self-correction manoeuvre to evaluate the pathology and pain-induced effects on NSLBP patients’ postural control compared to healthy young adults. In the performance of such an ISCO manoeuvre, it is difficult to distinguish between the contribution of sensory perception and motor control ability. Fortunately, the literature helps to clear doubts. Indeed, it was established that proprioception provides the most significant contribution to the perception of body orientation and related changes. For example, it is reported that, in a well-lit environment with a firm base of support, healthy individuals rely on a combination of somatosensory (70%), vision (10%), and vestibular (20%) information in order to maintain their upright posture [63]. In fact, although any part of the body surface can influence the control and perception of body orientation [62,64], evidence suggests that the representation of the body’s static and dynamic geometry may be primarily based on muscle proprioceptive inputs, which continuously inform the central nervous system about the position of each part of the body in relation to the others [62,65,66,67]. Moreover, with increasing stance width, lateral body motion is detected more easily by proprioceptors and less readily by vision or the vestibular system [68]. Worth noting, we asked the patients to align heels on a line parallel to the frontal plane and keep feet apart at about pelvis width, without restricting feet directions.

As for healthy young adults, our data confirm that perceiving and controlling posture “properly” in a global way is also very hard for NSLBP patients. In fact, in our sample population, only 15% of males and 4.7% of females could obtain a GPI “*improvement*”. Conversely, these results confirm that it is not an impossible task, but excellent proprioception and motor control are needed. On the other hand, the fact that participants demonstrated being able to focus only on few aspects at a time leads to additional consideration. For example, the changes detected for the |ASO FR| or |AGO FR| parameters in the frontal plane, during the ISCO manoeuvre, express if the subject can perceive and correct (with a voluntary motor control action) the eventual side overhanging of the trunk or the entire body, with respect to the gravitational line. NSLBP patients demonstrated slightly better performance than healthy young adults in either gender, aligning the trunk and the body with the gravity line in a better way, even if without a statistically significant difference (Table 6), albeit only a small percentage with respect to the total. Changing the amount of lateral leaning of the trunk or the body is an effortless motor control task, and it is easy to align both the trunk and the body along the gravitational line by adding visual feedback, especially if the pain is not elicited by such a manoeuvre. Such evidence can be trivially tested using a mirror and a plumb line. Therefore, it is unlikely to refer to a reduced capability in motor control by the relatively poor result on improving the posture connected to |ASO FR| or |AGO FR| parameters. This thesis is supported by the literature on proprioception and visual perception. Darling and Hondzinski [69] established that “there is a strong visual perceptual preference for the extrinsic gravitational axis over intrinsic axes fixed to body segments, and that the availability of visual background information does not improve perceptions of verticality”. More recently, Barbieri et al. [70], studying the modulation of proprioception by appropriate tendinomuscular vibration at Achilles tendons, demonstrated that proprioception contributes to the representation of the intrinsic body vertical. In our experiment, when the subject was asked to improve their posture, they did not have any other visual input than the blank wall visual background information. Thus, given the above-described information, the interpretation of the relatively poor result on the improvement of |ASO FR| or |AGO FR| parameters provided by NSLBP patients and by healthy young adults [26] is more likely due to a difficulty/lack of accuracy in the perception/proprioception of the gravity line direction than a reduced capability in motor control.

Detailing the differences in NSLBP patients vs. healthy young adults, some common behaviour and some gender specificity can be observed. As evaluated through the |∆PSIS| parameter, the pelvis obliquity resulted in being consistently greater in NSLBP patients vs. healthy young adults for both sexes and both IO and ISCO conditions, showing a structural difference between the healthy and the pathologic group. Such an occurrence was also found in D’Amico et al. (2021) [31]. In males, the pelvis obliquity was also associated with greater |∆ASIS| and underfoot load discrepancy (|∆UL|) compared to healthy young adults. Lastly, NSLBP patients and healthy young adults presented the same pelvis torsion |PT| for both genders. In their study, D’Amico et al. [25] found in healthy young adults a direct connection between pelvis tilt/torsion and leg length discrepancy (LLD), in that, when correcting the LLD using an underfoot wedge of appropriate thickness, the pelvis approached horizontality. The same result was obtained by D’Amico et al. (2021) [31], investigating the effects of LLD equalisation in the immediate, medium, and long term in a population of NSLBP subjects. Thus, the above results suggest a higher LLD in NSLBP patients because their pelvis tilt is greater than that of healthy young adults. Evidence between LBP and LLD connection also appears in other literature studies [71,72,73,74], even if such a topic represents an open debate [75].

Spine deformities (measured by the CA1 and CA2 parameters) resulted greater in NSLBP patients than in healthy young adults for both sexes and both IO and ISCO conditions, even if not always statistically significant. The NSLBP sample did not show any ISCO vs. IO spine deformities changes, while healthy young adults showed slight improvement. In such a case, it is impossible to establish if the reduced capability of spine deformities reduction in NSLBP patients is related to more structured deformities or a lesser proprioception and motor control ability. Conversely, D’Amico et al. (2021) [31] found that, in NSLBP patients having LLD, spine deformities were significantly reduced together with other unbalanced postural parameters, after an adaptation period, by adopting customised foot orthotics with corrective heel lift that led to a complete remission of pain. Such a reduction in spine deformities was not observed in the immediate term when heel lift correction was applied. From such results, it can be argued that back pain could be connected to structured spine deformities. Worth noting, the frontal plane principal spine deformity (CA1) was located in the thoracolumbar and lumbar regions for about two-thirds of the patients. The literature observed a correlation between LLD and the presence of spine deformities, especially in the lumbar region [72,73,76,77,78,79,80]. Hence, it could be hypothesised that the more significant spinal deformities observed in the NSLBP sample than healthy young adults could be correlated to the higher pelvis obliquity found in such a group.

In the sagittal plane, NSLBP males showed a backward global leaning posture in both IO and ISCO compared to healthy young adults, as witnessed by lower AGO SG, ASO SG, and SA parameters. Conversely, no difference was found for females. Compared to men, the literature reports that women are characterised by enlarged peripheral fat depots (gluteo-femoral region), whereas intra-abdominal fat depots are preferentially increased in men [81]. Our NSLBP male sample confirmed a more significant mass at the abdominal level in the body’s anterior aspect than healthy young adults. A possible biomechanical explanation of the backward leaning found in males could be a compensated posture to counteract the greater belly prominence. The performance of the ISCO manoeuvre by the NSLBP sample showed the same proprioception and motor control ability found in healthy young adults, i.e., a flattening of thoracic kyphosis (TKA), an unmovable lumbar lordosis (LLA), an increase in sacral angle (SA), and a general tendency to increase forward leaning of the trunk. Interestingly, the same behaviour in the sagittal plane postural modification during ISCO was recently observed in adolescent idiopathic scoliosis patients [27].

The intra-subject analysis confirmed the substantially equivalent behaviour of NSLBP patients and healthy young adults. For both populations, the better focus on proprioception and motor control was in the sagittal plane. Most NSLBP patients were unable to change their posture globally in ISCO (80% males and 93% females were classified as unchanged). The NSLBP group (both genders) presented a sort of lumbar neglect, as found in the healthy young adults, as almost two-thirds of the patients showed no change in LLA values. In the small percentage of patients showing changes in ISCO, males performed better than females either globally (GPI, 15% vs. 4.7%) or in the performances grouped by planes (SPI, 27.5% vs. 14.0%) (FPI, 17.5% vs. 11.6%), while females performed better only in the underfoot load rebalancing (|∆UL|, 40.0% vs. 35.1%). Altogether, patients with NSLBP showed a better ability than healthy young adults to reduce underfoot load unbalancing. Such improved performance could be linked to the more significant underfoot load asymmetry in the IO found in patients with NSLBP compared to healthy young adults. Indeed, it can be argued that patients with NSLBP might better perceive their greater load asymmetries, trying to equalize them when the ISCO manoeuvre is required.

Definitely, intra-subject analysis confirmed that, when changes occurred, NSLBP patients could not focus and control their posture globally, but they could focus only on a few aspects at a time, individually. This was also demonstrated for healthy young adults [26] and very recently for adolescent idiopathic scoliotic patients [27].

The best values of improvement were obtained in the male group in |ASO FR| (42.5%), SA (47.5%), TKA (37.5%), and |∆PSIS| (35.0%) and in the female group in |∆UL| (40.0%), |ASO FR| (37.2%), and SA (34.9%). However, even worsening scored high in some parameters. Curiously, both genders presented a worse percentage in the following two parameters: thoracic kyphosis (TKA: 35.0% males, 35.3% females) and pelvis torsion (|PT|: 37.5% males, 34.9% females). Considering the summarising indices (FPI, SPI, GPI), males showed higher percentages in terms of both improvement and worsening. As for healthy young adults, the percentage of improved/worsened summarising indices (FPI, SPI, GPI) resulted in general far below that obtained for every single postural parameter. The same was confirmed for adolescent idiopathic scoliosis patients [27]. The lumbar level showed the largest unmodified behaviour. Laird et al. [14], in their review, showed a reduced lumbar range of movement and proprioception in a population with LBP. Given the results obtained in previous studies [26,27], regardless of sex and healthy or pathological condition, subjects presented an individually managed poor proprioception and postural control capacity both globally and in the trunk. Furthermore, the instinctive self-correction manoeuvre induced similar postural reactions in all groups with greater involvement in the thoracic tract and neglect of the lumbar tract.

Physical rehabilitation aims to restore the function of specific muscles and improve the postural performance by stimulating, via proper motor exercises, the appropriate integration and managing in all the central nervous system components controlled posture, with a focus on trunk and spine [16,82]. NSLBP is multifactorial [83,84], and no single exercise programme is optimal for all NSLBP patients [85]. However, MCE effectiveness compared to other approaches is debated [16,23,24]. The inconsistency of results is likely explained by studies taking a one-size-fits-all approach, even though the literature shows that patients’ characteristics vary together with their motor control pain-induced modifications [21]. Patient subgrouping was hypothesised as a possible solution toward the personalisation of exercise treatment [21,86,87]. However, the currently used clinical-based approaches do not reach a unanimous consensus on the best way to subgroup patients [88,89].

Our results indicate that the individualisation of therapy is necessary to reach treatment efficacy.

The optoelectronic stereophotogrammetric approach we used in this research demonstrated its ability to quantitatively describe the 3D standing posture of each patient, and the ISCO manoeuvre helped to characterise the patients’ proprioception and motor control. D’Amico et al. (1995) [38] were the first to explore the potential of such a methodology to accurately reconstruct and quantify the 3D spine structure. The consensus between the optoelectronic stereophotogrammetric spine shape measurements and X-ray assessment was shown in an in vivo comparative study on scoliotic patients, where X-ray and stereophotogrammetric evaluations were conducted within minutes of each other [51]. More recently, such a method was used to determine the baseline reference normative data of a healthy young adult population, demonstrating a high agreement with results obtained via other methods as presented in the existing literature. In particular, the optoelectronic stereophotogrammetric approach demonstrated definite consistency with normative data results obtained from X-ray measurements (especially in the spine shape measurements), which are currently the “gold standard” [25,33].

The optoelectronic stereophotogrammetric measurement protocol presented here resulted in being detailed and fast enough to represent a helpful tool for future use in sub-grouping and individualising therapy plans. The entire measurement session, including posture and ISCO analysis, takes less than 30 min from patient entry to full report presentation.

According to the present study’s results, the detected transversal scarce proprioception involving healthy and pathologic subjects promotes an individualised MCE approach in NSLBP rehabilitation, possibly driven by 3D complete posture evaluation. Furthermore, proprioception is involved in the age declining process. A recent study highlighted a potential age-related deterioration in the central and peripheral processing of lumbar proprioceptive signals [90]. For such a reason, MCE could likely also play an important role in the prevention of NSLBP. An interesting study could be to evaluate in the long–medium term if a preventative individualised MCE training approach for healthy young adults could decrease the risk of NSLBP in the older population.

A limitation of this study is that the NSLBP group was slightly older than healthy young adults. However, we reasonably believe that such a slight age difference did not influence the outcomes in that they appeared more correlated with NSLBP postural characteristics. Nevertheless, there are inherent limitations to using cutaneous markers (such as those used here) in different kinds of populations. For example, in cases where there is voluminous subcutaneous tissue, due to the distance between the skin marker and the reference bones on the spine, the surface profile may diverge from the actual shape of the spine.

## 5. Conclusions

In conclusion, except for some essential anatomical–structural differences found in the subacute and chronic NSLBP sample, the patients did not show impaired proprioception and motor control capacity compared to healthy young adults. As for a healthy population, it was confirmed that the self-correcting manoeuvre that leads to a global improvement of the natural standing posture is not instinctive, and a specific postural training has to be planned to assimilate how to reach it. Like healthy young adults and adolescents with idiopathic scoliosis, patients with subacute and chronic NSLBP fail to focus and control their posture globally but can only focus on a few aspects at a time, on an individual basis. Participants showed better “attention” to the sagittal plane but with substantial neglect of the lumbar spine. Postural changes from the ISCO manoeuvre rarely induced a better posture than IO. A learned self-corrected posture, with specific postural training, will be expected to improve proprioception and motor control capacity and reduce pain in NSLBP chronic patients. Overall, 3D stereophotogrammetry proved to be a helpful tool for quantitatively assessing the posture of the entire skeleton and deformities of the spine.

## Figures and Tables

**Figure 1 jcm-11-00546-f001:**
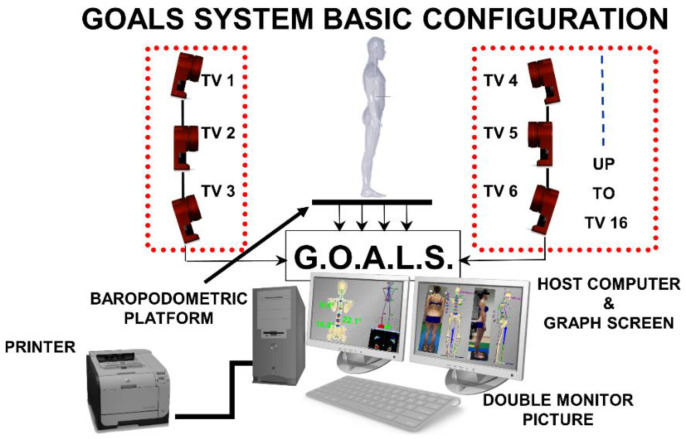
The experimental setup used for 3D posture analysis: GOALS system and baropodometric platform configuration.

**Figure 2 jcm-11-00546-f002:**
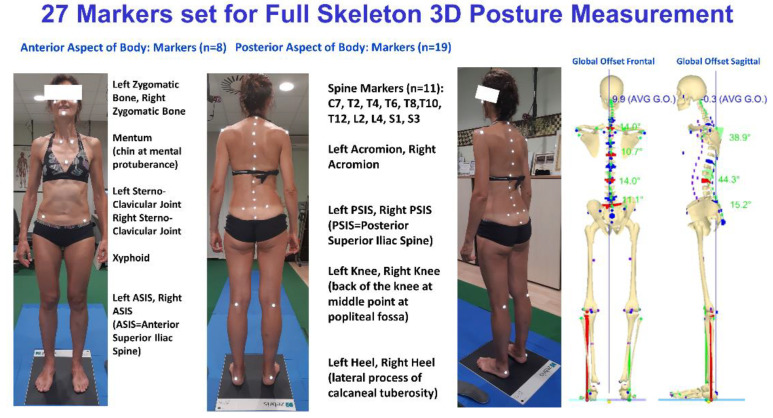
Example of marker sets for posture and full skeleton reconstruction, including the representation of vertical forces.

**Figure 3 jcm-11-00546-f003:**
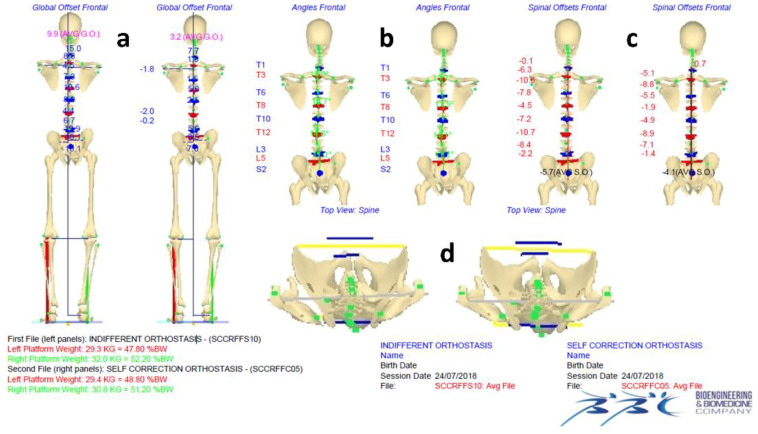
Example of data elaboration outcome and the related graphical report of the IO vs. ISCO full skeleton reconstructions comparison in the frontal (**a**–**c**) and horizontal (**d**) planes, including a representation of the vertical forces.

**Figure 4 jcm-11-00546-f004:**
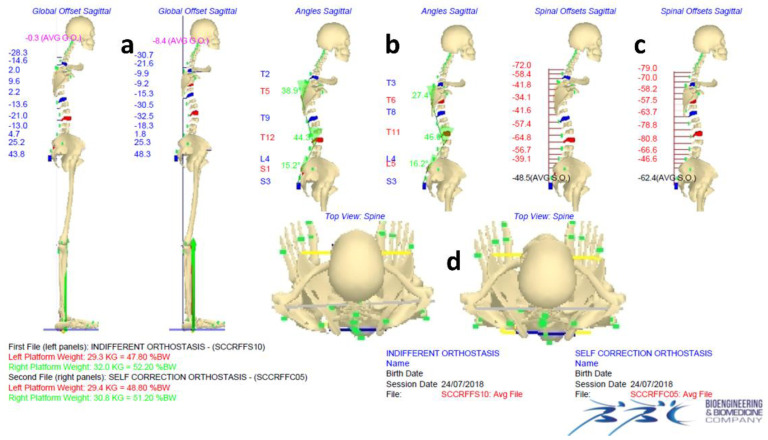
Example of data elaboration outcome and the related graphical report of the comparison of IO vs. ISCO full skeleton reconstructions in the sagittal (**a**–**c**) and horizontal planes (**d**), including a representation of the vertical forces.

**Table 1 jcm-11-00546-t001:** Sample population characteristics: total 83 NSLBP patients.

Population Characteristics	Females (*n* = 43)		Males (*n* = 40)		*t*-Test Females vs. Males
	Range	Mean (SD)	Range	Mean (SD)	
Age (years) *	21–40	31.2 ± 5.6	21–40	30.8 ± 5.1	ns
Height (cm)	149–180	163.1 ± 6.0	154–193	176.4 ± 8.7	*p* = 2.0 × 10^−11^
Weight (kg)	47–89	62.0 ± 10.7	54–108	77.1 ± 12.2	*p* = 7.4 × 10^−8^
BMI (kg/m^2^)	17.7–33.1	23.4 ± 4.3	20.5–33.3	24.7 ± 2.8	ns

* Total NSLBP patients mean age =31 ± 5.3; ns = not significant.

**Table 2 jcm-11-00546-t002:** NSLBP patient characteristics compared to healthy young adult characteristics.

	NSLBP Mean (SD)	HYAP Mean (SD)	*t*-Test
Age males (years)	30.8 ± 5.1	24.9 ± 3.9	*p* = 3.5 × 10^−8^
Age females (years)	31.2 ± 5.6	23.5 ± 3.2	*p* = 3.6 × 10^−11^
Weight males (kg)	77.1 ± 12.2	73.9 ± 9.3	ns
Weight females (kg)	62.0 ± 10.7	57.7 ± 9.1	*p* = 0.036
Height males (cm)	176.4 ± 8.7	178.3 ± 6.5	ns
Height females (cm)	163.1 ± 6.0	164.3 ± 5.3	ns
BMI males (kg/m^2^)	24.7 ± 2.8	23.2 ± 2.1	*p* = 0.0056
BMI females (kg/m^2^)	23.4 ± 4.3	21.3 ± 2.6	*p* = 0.0065

ns = not significant.

**Table 3 jcm-11-00546-t003:** List of considered parameters (definitions and corresponding acronyms) for IO vs. ISCO comparison and summarising indices.

Global Summarising Index	Parameters			Specific Summarising Indices
**GPI** **Global postural index**	**Acronyms**	**Descriptions**	**Definitions**	
	|ASO FR| (mm)	|Average frontal spinal offsets|	The ASO is the mean of the horizontal distances in the frontal plane of each labelled spine landmark with respect to the vertical axis passing by S3; absolute value of the average to disregard the side	**FPI** **Frontal postural index**
	|AGO FR| (mm)	|Average frontal global offsets|	The AGO is the mean of the horizontal distances in the frontal plane of each labelled spine landmark respect to the vertical axis passing through the middle point between heels; absolute value of the average to disregard the side	
	|∆ASIS| (mm)	|∆Anterior superior iliac spine|	Absolute ASIS height difference in frontal plane	
	|∆PSIS| (mm)	|∆Posterior superior iliac spine|	Absolute PSIS height difference in frontal plane	
	CA1; CA2 (°)	1° Cobb angle; 2° Cobb angles	Cobb angles of the two main “spinal deformities” found in the frontal plane	
	|PT (mm)|	|Pelvis torsion| = |(∆ASIS − ∆PSIS)|	Rotation of the right respect to the left innominate bone. Rotations are intended around a horizontal axis running through the symphysis pubis. Absolute value to disregard the side	**SPI** **Sagittal postural index**
	ASO SG (mm)	Average sagittal spinal offsets	The ASO SG is the mean of horizontal distances in the sagittal plane of each labelled spine landmark respect to the vertical axis passing by S3; negative values represent forward leaning	
	AGO SG (mm)	Average sagittal spinal offsets	The AGO SG is the mean of horizontal distances in the sagittal plane of each labelled spine landmark respect to the vertical axis passing through the middle point between heels; negative values represent forward leaning	
	SA (°)	Sacral angle	The inclination of the S1–S3 line with respect to the vertical line	
	TKA (°)	“Thoracic” kyphosis angles	Kyphosis and lordosis are correctly identified following spine curvature spatial changes at inflexion points; thus, limit vertebrae are not strictly bounded to the specific anatomical region	
	LLA (°)	“Lumbar” lordosis angles		
	|∆UL| (%BW)	|∆Underfoot load|	Left vs. right side body weight (bw) percentage difference; absolute value to disregard the side	

**Table 4 jcm-11-00546-t004:** NSLBP male vs. female comparisons in both IO and ISCO: Hotelling T^2^ tests results, 95% confidence intervals, and difference of means.

	Hotelling T^2^ Test for Independent Samples NSLBP Male: vs. Female in IO and ISCO Comparison								
		IO (*n*1 = 40, *n*2 = 43, *k* = 13, *T2* = 64.3, *p* = 3.9 × 10^−5^, *d* = 1.76, Power = 0.99)				ISCO (*n*1 = 40, *n*2 = 43, *k* = 13, *T2* = 78.7, *p* = 2.6 × 10^−6^, *d* = 1.94, Power = 0.99)			
Parameter	Descriptions	Males Mean	Females Mean	Difference in Means	CI 95% Lower, Upper	Males Mean	Females Mean	Difference in Means	CI 95% Lower, Upper
|ASO FR| (mm)	|Average frontal spinal offsets|	7.3 ± 4.9	5.9 ± 3.9	1.47	−0.93, 3.07	6.6 ± 5.6	5.4 ± 3.9	1.15	−0.94, 3.25
|AGO FR| (mm)	|Average frontal global offsets|	12.1 ± 11.1	8.4 ± 6.1	3.61	−2.13, 4.41	10.9 ± 9.2	8.4 ± 5.9	2.43	−0.92, 5.78
CA1 (°)	1° Cobb angle;	13.1 ± 8.1	13.4 ± 6.4	−0.28	−3.92, 1.35	13.1 ± 7.6	12.6 ± 6.2	0.42	−2.61, 3.46
CA2 (°)	2° Cobb angles	8.4 ± 6.9	9.2 ± 5.9	−0.76	−4.79, −0.19	9.4 ± 6.9	9.2 ± 5.2	0.19	−2.47, 2.85
TKA (°)	“Thoracic” kyphosis angles	48.2 ± 12.0	45.5 ± 10.4	2.71	0.54, 7.45	39.4 ± 13.0	37.6 ± 10.3	1.83	−3.27, 6.93
LLA (°)	“Lumbar” lordosis angles	33.5 ± 9.3	41.4 ± 8.8	**−7.87 ***	−6.49, −0.23	34.6 ± 9.5	42.1 ± 9.0	**−7.49 ***	−11.52, −3.45
|∆ASIS| (mm)	|∆Anterior superior iliac spine|	10.5 ± 7.9	7.8 ± 5.0	2.76	−0.10, 4.68	10.6 ± 8.2	7.6 ± 4.7	**3.05 ***	0.16, 5.94
|∆PSIS| (mm)	|∆Posterior superior iliac spine|	7.5 ± 4.0	6.2 ± 3.2	1.29	−0.60, 2.43	7.5 ± 4.4	6.0 ± 3.1	1.53	−0.14, 3.19
|PT| (mm)	|Pelvis torsion| = |(∆ASIS − ∆PSIS)|	6.3 ± 4.5	6.5 ± 5.5	−0.25	0.13, 3.28	6.8 ± 4.7	6.4 ± 5.7	0.42	−1.86, 2.71
SA (°)	Sacral angle	13.4 ± 6.3	17.4 ± 6.6	**−4.00 ***	−4.22, −0.03	15.7 ± 6.4	19.1 ± 6.1	**−3.32 ***	−6.05, −0.59
ASO SG (mm)	Average sagittal spinal offsets	−8.2 ± 16.0	−21.5 ± 15.0	**13.30 ***	5.75, 14.56	−11.2 ± 12.5	−25.6 ± 13.9	**14.36 ***	8.58, 20.14
AGO SG (mm)	Average sagittal global offsets	3.3 ± 24.4	7.5 ± 26.0	−4.22	−14.81, −0.24	6.8 ± 24.0	9.8 ± 23.7	−3.02	−13.44, 7.41
|∆UL| (%BW)	|∆Underfoot load|	8.1 ± 6.2	5.4 ± 4.6	**2.68 ***	−1.46, 1.64	7.3 ± 6.1	5.5 ± 4.1	1.83	−0.44, 4.09

***** Bold numbers indicate a statistically significant difference of means.

**Table 5 jcm-11-00546-t005:** IO vs. ISCO comparisons by gender in NSLBP patients: Hotelling T^2^ tests results, 95% confidence intervals, and difference of means.

	Hotelling T^2^ Test for Paired Samples: per Gender IO vs. ISCO Comparison								
		Males (*n* = 40, *k* = 13, *T2* = 347.3, *p* = 3.8 × 10^−10^, *d* = 2.94, Power = 0.99)				Females (*n* = 43, *k* = 13, *T2*= 82.3, *p* = 3.1 × 10^−4^, *d* = 1.38, Power = 1.0)			
Parameter	Descriptions	IO Mean	ISCO Mean	Difference in Means	CI 95% Lower, Upper	IO Mean	ISCO Mean	Difference in Means	CI 95% Lower, Upper
|ASO FR| (mm)	|Average frontal spinal offsets|	7.3 ± 4.9	6.6 ± 5.6	0.78	−0.67, 2.23	5.9 ± 3.9	5.4 ± 3.9	0.46	−0.32, 1.25
|AGO FR| (mm)	|Average frontal global offsets|	12.1 ± 11.1	10.9 ± 9.2	1.19	−2.02, 4.41	8.4 ± 6.1	8.4 ± 5.9	0.02	−1.68, 1.71
CA1 (°)	1° Cobb angle;	13.1 ± 8.1	13.1 ± 7.6	0.06	−1.45, 1.58	13.4 ± 6.4	12.6 ± 6.2	0.77	−1.22, 2.75
CA2 (°)	2° Cobb angles	8.4 ± 6.9	9.4 ± 6.9	−0.97	−2.60, 0.66	9.2 ± 5.9	9.2 ± 5.2	−0.01	−1.24, 1.22
TKA (°)	“Thoracic” kyphosis angles	48.2 ± 12.0	39.4 ± 13.0	**8.82 ***	6.32, 11.32	45.5 ± 10.4	37.6 ± 10.3	**7.94 ***	4.10, 11.78
LLA (°)	“Lumbar” lordosis angles	33.5 ± 9.3	34.6 ± 9.5	−1.06	−3.12, 1.00	41.4 ± 8.8	42.1 ± 9.0	-0.68	−3.00, 1.65
|∆ASIS| (mm)	|∆Anterior superior iliac spine|	10.5 ± 7.9	10.6 ± 8.2	−0.09	−1.10, 0.92	7.8 ± 5.0	7.6 ± 4.7	0.19	−0.52, 0.91
|∆PSIS| (mm)	|∆Posterior superior iliac spine|	7.5 ± 4.0	7.5 ± 4.4	−0.01	−0.49, 0.46	6.2 ± 3.2	6.0 ± 3.1	0.22	−0.02, 0.46
|PT| (mm)	|Pelvis torsion| = |(∆ASIS − ∆PSIS)|	6.3 ± 4.5	6.8 ± 4.7	−0.53	−1.39, 0.32	6.5 ± 5.5	6.4 ± 5.7	0.14	−0.57, 0.85
SA (°)	Sacral angle	13.4 ± 6.3	15.7 ± 6.4	**−2.38 ***	−3.17, -1.59	17.4 ± 6.6	19.1 ± 6.1	**−1.71 ***	−2.71, -0.70
ASO SG (mm)	Average sagittal spinal offsets	−8.2 ± 16.0	-11.2 ± 12.5	3.05	−1.30, 7.40	−21.5 ± 15.0	−25.6 ± 13.9	**4.11 ***	1.18, 7.04
AGO SG (mm)	Average sagittal global offsets	3.3 ± 24.4	6.8 ± 24.0	−3.46	−8.28, 1.36	7.5 ± 26.0	9.8 ± 23.7	−2.26	−7.71, 3.20
|∆UL| (%BW)	|∆Underfoot load|	8.1 ± 6.2	7.3 ± 6.1	0.76	1.32, 2.84	5.4 ± 4.6	5.5 ± 4.1	−0.09	−1.83, 1.65

***** Bold numbers indicate a statistically significant difference of means.

**Table 6 jcm-11-00546-t006:** NSLBP patients vs. healthy young adults by gender comparisons in both IO and ISCO: Hotelling T^2^ tests results, 95% confidence intervals, and difference of means.

	Hotelling T^2^ Test for Independent Samples: NSLBP vs. HEALTHY YOUNG ADULTS in IO and ISCO Comparison								
		Females							
		IO (*n*1 = 43, *n*2 = 57, *k* = 13, *T2* = 43.9, *p* = 6.0 × 10^−4^, *d* = 1.16, Power = 0.99)				ISCO (*n*1 = 43, *n*2 = 57, *k* = 13, *T2* = 54.1, *p* = 1.3 × 10^−4^, *d* = 1.48, Power = 0.99)			
Parameter	Descriptions	NSLBP Mean	Healthy Young Adults Mean	Difference in Means	CI 95% Lower, Upper	NSLBPP Mean	Healthy Young Adults Mean	Difference in Means	CI 95% Lower, Upper
|ASO FR| (mm)	|Average frontal spinal offsets|	5.9 ± 3.9	6.5 ± 4.6	−0.69	−2.40, 1.03	5.4 ± 3.9	6.3 ± 4.1	−0.93	−2.53, 0.68
|AGO FR| (mm)	|Average frontal global offsets|	8.4 ± 6.1	12.1 ± 8.1	**−3.63 ***	−6.56, −0.69	8.4 ± 5.9	11.0 ± 8.1	−2.61	−5.53, 0.30
CA1 (°)	1° Cobb angle;	13.4 ± 6.4	10.3 ± 5.0	**3.12 ***	0.86, 5.37	12.6 ± 6.2	9.5 ± 4.8	**3.19 ***	1.00, 5.38
CA2 (°)	2° Cobb angles	9.2 ± 5.9	7.5 ± 4.1	1.73	−0.25, 3.70	9.2 ± 5.2	7.2 ± 3.9	**2.04 ***	0.22, 3.86
TKA (°)	“Thoracic” kyphosis angles	45.5 ± 10.4	47.2 ± 8.6	−1.71	−5.49, 2.07	37.6 ± 10.3	40.8 ± 8.7	−3.28	−7.06, 0.49
LLA (°)	“Lumbar” lordosis angles	41.4 ± 8.8	44.2 ± 9.7	−2.80	−6.52, 0.93	42.1 ± 9.0	43.7 ± 10.4	−1.63	−5.58, 2.31
|∆ASIS| (mm)	|∆Anterior superior iliac spine|	7.8 ± 5.0	8.2 ± 5.5	−0.45	−2.58, 1.67	7.6 ± 4.7	8.0 ± 5.6	−0.48	−2.58, 1.62
|∆PSIS| (mm)	|∆Posterior superior iliac spine|	6.2 ± 3.2	4.8 ± 2.6	**1.46 ***	0.32, 2.60	6.0 ± 3.1	4.7 ± 2.6	**1.28 ***	0.15, 2.41
|PT| (mm)	|Pelvis torsion| = |(∆ASIS − ∆PSIS)|	6.5 ± 5.5	5.45 ± 3.9	1.05	−0.82, 2.92	6.4 ± 5.7	5.6 ± 4.4	0.80	−1.10, 2.71
SA (°)	Sacral angle	17.4 ± 6.6	17.3 ± 5.9	0.08	−2.40, 2.56	19.1 ± 6.1	18.2 ± 5.0	0.83	−1.38, 3.03
ASO SG (mm)	Average sagittal spinal offsets	−21.5 ± 15.0	−20.6 ± 11.9	−0.85	−6.20, 4.50	−25.6 ± 13.9	−23.5 ± 11.6	−2.09	−7.15, 2.97
AGO SG (mm)	Average sagittal global offsets	7.5 ± 26.0	−1.8 ± 26.7	9.32	−1.26, 19.89	9.8 ± 23.7	−0.4 ± 26.9	10.18	−0.08, 20.45
|∆UL| (%BW)	|∆Underfoot load|	5.4 ± 4.6	5.1 ± 4.3	0.33	−1.44, 2.10	5.5 ± 4.1	5.4 ± 3.7	0.13	−1.43, 1.70
		**Males**							
		**IO (*n*1 = 40, *n*2 = 64, *k* = 13, *T2* = 67.7, *p* = 5.4****× 10^−6^**, ***d* = 1.65, Power = 0.99)**				**ISCO (*n*1 = 40, *n*2 = 64, *k* = 13, *T2* = 56.4, *p* = 6.7****× 10^−5^**, ***d* = 1.51, Power = 0.99)**			
**Parameter**	**Descriptions**	**NSLBP Mean**	**Healthy Young Adults Mean**	**Difference in Means**	**CI 95%** **Lower, Upper**	**NSLBP Mean**	**Healthy Young Adults Mean**	**Difference in Means**	**CI 95%** **Lower, Upper**
|ASO FR| (mm)	|Average frontal spinal offsets|	7.3 ± 4.9	6.2 ± 5.1	1.09	−0.92, 3.10	6.6 ± 5.6	5.8 ± 4.6	0.73	−1.26, 2.72
|AGO FR| (mm)	|Average frontal global offsets|	12.1 ± 11.1	11.6 ± 8.4	0.45	−3.36, 4.26	10.9 ± 9.2	12.8 ± 8.7	−1.90	−5.45, 1.65
CA1 (°)	1° Cobb angle;	13.1 ± 8.1	11.5 ± 5.4	1.68	−0.94, 4.30	13.1 ± 7.6	10.4 ± 5.3	**2.69 ***	0.16, 5.22
CA2 (°)	2° Cobb angles	8.4 ± 6.9	7.2 ± 4.3	1.18	−0.99, 3.36	9.4 ± 6.9	7.0 ± 4.7	**2.39 ***	0.14, 4.65
TKA (°)	“Thoracic” kyphosis angles	48.2 ± 12.0	45.1 ± 8.9	3.13	−0.94, 7.20	39.4 ± 13.0	36.4 ± 8.4	2.96	−1.19, 7.11
LLA (°)	“Lumbar” lordosis angles	33.5 ± 9.3	32.6 ± 8.1	0.87	−2.57, 4.30	34.6 ± 9.5	32.3 ± 8.4	2.27	−1.26, 5.81
|∆ASIS| (mm)	|∆Anterior superior iliac spine|	10.5 ± 7.9	7.5 ± 5.3	**3.00 ***	0.43, 5.56	10.6 ± 8.2	7.6 ± 5.2	**2.98 ***	0.38, 5.57
|∆PSIS| (mm)	|∆Posterior superior iliac spine|	7.5 ± 4.0	5.1 ± 2.2	**2.43 ***	1.22, 3.65	7.5 ± 4.4	5.1 ± 2.2	**2.42 ***	1.12, 3.72
|PT| (mm)	|Pelvis torsion| = |(∆ASIS − ∆PSIS)|	6.3 ± 4.5	5.3 ± 4.5	0.84	−0.98, 2.65	6.8 ± 4.7	5.6 ± 4.8	1.15	−0.75, 3.05
SA (°)	Sacral angle	13.4 ± 6.3	15.7 ± 5.5	**−2.34 ***	−4.66, −0.02	15.7 ± 6.4	16.8 ± 5.5	−1.06	−3.40, 1.28
ASO SG (mm)	Average sagittal spinal offsets	−8.2 ± 16.0	−14.0 ± 12.4	**5.83 ***	0.29, 11.38	−11.2 ± 12.5	−17.4 ± 13.5	**6.22 ***	0.98, 11.46
AGO SG (mm)	Average sagittal global offsets	3.3 ± 24.4	−10.2 ± 21.5	**13.47 ***	4.42, 22.52	6.8 ± 24.0	−8.8 ± 19.4	**15.54 ***	7.03, 24.04
|∆UL| (%BW)	|∆Underfoot load|	8.1 ± 6.2	4.5 ± 3.8	**3.55 ***	1.61, 5.50	7.3 ± 6.1	5.1 ± 4.5	**2.25 ***	0.17, 4.32

***** Bold numbers indicate a statistically significant difference of means.

**Table 7 jcm-11-00546-t007:** Intra-subject statistical analysis.

		NSLBP Males			NSLBP Females			Healthy Young Adults		
3D Posture Parameter	Descriptions	*Improvement*	*Worsening*	*Unchanged*	*Improvement*	*Worsening*	*Unchanged*	*Improvement*	*Worsening*	*Unchanged*
|ASO FR|	|Average frontal spinal offsets|	42.5%	15.0%	42.5%	37.2%	23.3%	39.5%	29.8%	20.7%	49.6%
|AGO FR|	|Average frontal global offsets|	25.0%	17.5%	57.5%	7.0%	23.3%	69.8%	26.4%	30.6%	43.0%
|∆ASIS|	|∆Anterior superior iliac spine|	32.5%	32.5%	35.0%	30.2%	20.9%	48.8%	19.8%	14.0%	66.1%
|∆PSIS|	|∆Posterior superior iliac spine|	35.0%	22.5%	42.5%	25.6%	14.0%	60.5%	21.5%	19.0%	59.5%
CA1	1° Cobb angle	30.0%	25.0%	45.0%	32.6%	11.6%	55.8%	28.1%	23.1%	48.8%
CA2	2° Cobb angles	5.0%	35.0%	60.0%	27.9%	20.9%	51.2%	25.6%	26.4%	47.9%
|PT|	|Pelvis torsion| = |(∆ASIS − ∆PSIS)|	30.0%	37.5%	32.5%	32.6%	34.9%	32.6%	29.8%	35.5%	34.7%
SA	Sacral angle	47.5%	20.0%	32.5%	34.9%	9.4%	55.8%	35.5%	5.8%	58.7%
TKA	“Thoracic” kyphosis angles	37.5%	35.0%	27.5%	25.6%	35.3%	39.5%	36.4%	27.3%	36.4%
LLA	“Lumbar” lordosis angles	25.0%	12.5%	62.5%	20.9%	16.5%	62.8%	20.7%	12.4%	66.9%
|∆UL|	|∆Underfoot load average|	35.1%	27.0%	37.8%	40.0%	22.5%	37.5%	22.5%	27.5%	50.0%
FPI	Frontal postural index	17.5%	17.5%	65.0%	11.6%	7.0%	81.4%	14.0%	9.9%	76.0%
SPI	Sagittal postural index	27.5%	15.0%	57.5%	14.0%	7.0%	79.1%	27.3%	10.7%	62.0%
GPI	Global postural index	15.0%	5.0%	80.0%	4.7%	2.3%	93.0%	6.6%	6.6%	86.8%

## Data Availability

The data presented in this study are available on request from the corresponding author.

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
