# Peer review of "3D Stereophotogrammetric Quantitative Evaluation of Posture and Spine Proprioception in Subacute and Chronic Nonspecific Low Back Pain"

_jcm, 2022, doi:10.3390/jcm11030546_

Round 1

Reviewer 1 Report

The manuscript is clear, not so relevant in Clinical Medicine Journal, but it presents a relevant theme: NSLBP.

It’s presented in a well-structured manner.

The manuscript sounds scientific, and the experimental design is well appropriated to the analysed issue.

Article is easy to interpret and understand, including details regarding the statistical analysis and data interpretation.

The manuscript’s results are reproducible based on the details given in the methods section.

The conclusions may be slightly improved, according to the evidence and arguments presented. I agree a self-improving of the standing posture is not instinctive but saying it “must be learned” is not a conclusion from your study. Maybe you can say: a learned self-corrected posture, with specific postural training, will be expected to improve a better proprioception and motor control capacity and reduce pain in NSLBP chronic patients.

Author Response

We thank Reviewer #1 for his work and suggestions. We agree with his/her note on the conclusions.

Thus we provide a revised version of our manuscript in which we modified the conclusions following the Reviewer #1 suggestion and now appear as the following (modifications are noted in bold characters and the sentence "must be learned" has been removed:

"In conclusion, except for some essential anatomical-structural differences found in the sub-acute and chronic NSLBP sample, the patients did not show impaired proprioception and motor control capacity compared to healthy young adults. As for a healthy population, it has been confirmed that the self-correcting manoeuvre that leads to a global improvement of the natural standing posture is not instinctive and (must be learned with) a specific postural training has to be planned to assimilate how to reach it. Like healthy young adults and adolescents with idiopathic scoliosis, patients with sub-acute and chronic NSLBP fail to focus and control their posture globally but can only focus on a few aspects at a time, on an individual basis. Participants showed better "attention" to the sagittal plane but with substantial neglect of the lumbar spine. Postural changes from the ISCO manoeuvre rarely induced a better posture than IO. A learned self-corrected posture, with specific postural training, will be expected to improve a better proprioception and motor control capacity and reduce pain in NSLBP chronic patients. 3D stereo-photogrammetry has proved to be a helpful tool for quantitatively assessing the posture of the entire skeleton and deformities of the spine."

Reviewer 2 Report

The study described in the publication "3D STereophotogrtammetric Quantitative Evaluation of Poasture and Spine Proproception in Nonspecific Low Back Pain (NLBP)" is aimed at the quantitative evaluation of whole skeletal posture and spinal deformities by 3D stereophotogrammetry in a series of NLBP patients compared to healthy controls. The data obtained from these observations were used to devise a program of self-correction maneuvers of the upright posture that resulted in objective improvement of symptoms in patients with NLBP.

The study is original and very well executed with a careful methodology of data collection and statistical processing. The text is detailed and although a bit long, it clearly presents the background of the study, the state of the art, the objectives pursued and the results obtained.

Author Response

We thank Reviewer #2 for his/her work and nice words for our manuscript.